# AI-powered detection of cyberbullying in short-form video content: A hybrid deep learning framework

Ahmad A. Mazhar[1], Islam Zada[2]*, Manal Aldhayan[3], Seetah Alsalamah[4], Mashael M. Asiri[5], Manel Ayadi[6], Abdullah Alshahrani[7]

1 College of Communication, University of Sharjah, Sharjah, United Arab Emirates, 2 Faculty of Computing & IT, Department of Software Engineering, International Islamic University Islamabad, Islamabad, Pakistan, 3 Computer Science Department College of Computer and IT, Shaqra University Shaqra, Shaqra, Saudi Arabia, 4 Department of Computer Science, College of Computer and Information Sciences, King Saud University, Riyadh, Saudi Arabia, 5 Technical and Engineering Specializations Unit, Applied College at Mahayil, King Khalid University, Abha, Saudi Arabia, 6 Department of Information Systems, College of Computer and Information Sciences, Princess Nourah bint Abdulrahman University, Riyadh, Saudi Arabia, 7 Department of Computer Science and Artificial Intelligence, College of Computer Science and Engineering, University of Jeddah, Jeddah, Saudi Arabia

* islam.zada@iiu.edu.pk

## Abstract

The explosive rise of short-form video platforms such as Instagram Reels, TikTok, and YouTube Shorts has transformed digital expression while intensifying the spread of cyberbullying. Unlike video abuse conveys multimodal cues visual, text-based harassment, auditory, and textual that challenge conventional detection methods. This study presents a hybrid deep-learning framework that integrates Convolutional Neural Networks (CNNs) for spatial features, Bidirectional Long Short Term Memory (BiLSTM) networks for temporal acoustic patterns, and a Transformer-based textual encoder to analyze synchronized video, audio, and caption streams. A semantic-consistency validation layer enforces cross-modal alignment using attention-based similarity constraints, ensuring that incongruent cues are penalized during classification. Experiments on two benchmark datasets, CAVD and SocialVidMix, demonstrate state-of-the-art performance accuracy 91.6%, precision 89.7%, recall 93.0%, and F1-score 91.3% with consistent results across Instagram, TikTok, and YouTube Shorts. The framework's, interpretability, robustness and scalability indicate strong potential for real-time deployment in automated content-moderation systems.

## 1 Introduction and motivation

The fast growth of short form video apps such as Tik Tok, Instagram Reels, and YouTube Shorts have changed the digital interaction across the globe [1]. These platforms promote creativity and social interaction but have also contributed to the proliferation of cyberbullying or online aggression that causes psychological damage

**Data availability statement:** All the relevant data are inside the text within this manuscript, and its Supporting Information files are in (S1 Data). Benchmark datasets used in this study such as: SocialVidMix, and CAVD are publicly available at the sources cited in References [62,63].

**Funding:** The author(s) received no specific funding for this work.

**Competing interests:** The authors have declared that no competing interests exist.

by being humiliating, embarrassing or organized into a systematic harassment. Video centered bullying combines gestures, voice, and word meaning in complex multi-modal patterns, unlike text-based abuse, which makes it difficult to identify using traditional methods. The methods used to detect early were largely based on keyword filtering and sentiment analysis, and it provided bare minimum context. Even trained deep-learning models trained on one modality like CNN-based visual classifiers or models based on transformers trained on text tend to fail when the presence of abusive intent is not uniformly spread across modalities. Some of the common weaknesses are misinterpretation of sarcasm, cross-modal inconsistency and imbalance in the dataset that diminish the accuracy of automated moderation tools. Moreover, the lack of real-time, platform-neutral solutions is an obstacle to the feasibility of the usage of existing models [2–5].

To address these drawbacks, this paper suggested a hybrid deep-learning model that used CNNs to extract spatial features, a Transformer-based encoder to extract textual semantics, and BiLSTM networks to analyze the acoustic temporal features. The key aspect of this architecture is the semantic consistency validation layer which is implemented through the use of attention based similarity constraints to match audio, video, and text representations. This mechanism makes the model interpret intent consistency in situations where signals are conflicting or culturally subtle consequently eliminating false detection in the real world settings..

The research gap addressed herein lies in the lack of models that unify multimodal fusion, semantic alignment, and efficiency suitable for live content moderation. While previous works explored one or two of these aspects independently, few approaches achieve all three within a deployable AI framework. Accordingly, this study carry out the following four main objectives:

1. Design a multimodal CNN-LSTM-Transformer architecture that integrates temporal, spatial, and semantic learning for short form videos.

2. Implement a semantic-consistency validation mechanism to enforce cross modal coherence through attention based alignment.

3. Evaluate performance on benchmark datasets (SocialVidMix and CAVD) and across multiple social media platforms (Instagram, TikTok, YouTube Shorts).

4. Validate statistical significance, robustness, and generalization for realistic, automated content-moderation systems.

   The Major contributions of this research are given bellow as:

- A novel multimodal framework integrating CNN, BiLSTM, and Transformer components with a semantic consistency layer.

- Introduction of an alignment driven validation-strategy that penalizes inter modal conflicts.

- A comprehensive evaluation with ablation studies, cross-platform generalization, and ROC/PR analyses.

- Insights into practical deployment, explainability, emphasizing scalability, and ethical moderation of online-content.

This study thus advances AI-powered cyberbullying-detection from isolated modality analysis to an integrated, context aware, and operationally viable paradigm for protecting users in modern social media ecosystems. The remainder of this paper is structured as follows:

Section 2: literature review articulated the related studies on visual/audio based, text-based, and multimodal approaches to cyberbullying detection, highlighting their limitations and research gaps. Section 3: research methodology design and presents the proposed hybrid framework, detailing the feature extraction models, multimodal fusion strategy, and semantic consistency validation mechanism. Section 4: experimental setup and results describes the experimental setup, evaluation metrics, and datasets used to assess the proposed model's performance. Section 5: Discussion discusses the empirical results, ablation studies, and the cross platform generalization analysis. Finally, Section 6: Conclusion and Future work concludes the study and outlines future research directions.

## 2  Literature review

Research on cyberbullying detection has evolved from single modality text classification toward sophisticated multimodal reasoning that integrates linguistic, visual, and acoustic signals [6]. Early text-only systems relied on lexical features, sentiment polarity, and n-gram representations [7,8]. Although these methods achieved reasonable precision on static comment datasets, they failed to capture sarcasm, irony, and implicit aggression, which are pervasive in short-form content. Traditional machine-learning models such as Naïve Bayes and Support Vector Machines were later enhanced with word embeddings to improve semantic understanding [9,10], however, they still lacked adaptability to linguistic drift and domain variability across platforms. Subsequent studies began exploiting deep contextual models, particularly transformer-based architectures like BERT and RoBERTa, to model abusive language [11,12]. These architectures improved contextual comprehension by encoding bidirectional dependencies within text but remained inherently limited to the textual channel. When users communicate through memes, emojis, or videos, text alone cannot accurately reflect the intended sentiment. This realization led to an increasing interest in multimodal learning, where text is complemented by visual and acoustic cues.

Initial multimodal approaches combined convolutional neural networks (CNNs) for image analysis with recurrent layers for temporal dependencies. For instance, visual-temporal pipelines using CNN–LSTM hybrids improved contextual sensitivity in user-generated videos [13,14] but struggled under low-resolution conditions and failed to resolve contradictions between modalities. Studies incorporating prosodic and tonal information through audio embeddings further expanded the scope of emotion detection [15–17], yet these systems produced high false-positive rates whenever expressive or humorous tones co-occurred with neutral text. In general, early multimodal research demonstrated that while adding modalities enhances recall, semantic misalignment between channels remains the principal challenge. To reduce such inconsistencies, attention-based fusion mechanisms were introduced to assign dynamic weights to modality contributions [18,19]. These attention models improved focus on informative cues and suppressed redundant features, but their fusion remained largely heuristic, relying on concatenation or linear projection rather than semantic correlation. Consequently, modality interactions were captured syntactically rather than semantically, causing instability [20] when cues conflicted for example, when a smiling face accompanies a negative caption.

In 2021–2022 [21–23], a new generation of cross-modal transformers emerged to learn aligned representations from paired modalities [24]. These systems leveraged self-attention across textual and visual streams, yielding improved performance in sarcasm and hate-speech detection. However, they required vast annotated data and incurred heavy computational overhead. Lightweight CNN–BERT pipelines were subsequently proposed to balance efficiency and performance [25,26], but they still lacked an explicit mechanism to ensure inter-modality agreement. Studies evaluating cross-platform generalization found that models trained on English-dominant datasets underperformed on multilingual corpora containing

transliterated or code-mixed expressions [27,28], reinforcing the need for semantic level harmonization independent of linguistic form.

By 2023, researchers began integrating emotion-aware adversarial learning and generative models into cyberbullying detection [29,30]. These frameworks synthesized challenging samples to enhance robustness against adversarial content and emotion manipulation. Despite promising gains, adversarial networks often introduced noise that degraded interpretability. Emotion-aware multimodal GANs captured affective dependencies across modalities [31] but did not constrain consistency between generated and original semantic features, occasionally amplifying bias. Recent studies have also explored graph-based and consistency-learning strategies to preserve relational context among modalities. Graph attention networks modeled inter-modal dependencies at node level, while cross-modal consistency learning penalized disagreement between paired features [32,33]. These designs substantially improved stability in dynamic social-media contexts. Nevertheless, most consistency modules optimized correlation rather than causation, meaning the model learned that modalities co-occur, not necessarily that they express the same intent. Consequently, even state-of-the-art systems misinterpreted satirical or performative behavior as aggression when contextual cues diverged.

Parallel progress occurred in fake-content and misinformation detection, which shares similar multimodal challenges. Research on deepfake and manipulated-media detection demonstrated that combining spatial and temporal features through residual and recurrent networks enhances sensitivity to subtle discrepancies [34–36]. These findings informed later cyberbullying frameworks by emphasizing the importance of temporal coherence and feature-level alignment across modalities. However, while deepfake detectors measure physical or physiological inconsistency, cyberbullying detection demands semantic interpretation of human intention—an aspect still underexplored in prior work.

A complementary line of research investigated domain adaptation and transfer learning to improve model generalization across platforms such as TikTok, Instagram, and YouTube Shorts [37–39]. Domain-adaptive transformers and feature-normalization layers helped reduce overfitting to specific platform styles [40], yet without semantic alignment, transferred models occasionally propagated cultural bias. Furthermore, most publicly available datasets are small and imbalanced, containing far fewer positive bullying samples than benign ones, which limits supervised learning performance. Meta-learning and few-shot strategies attempted to overcome this imbalance [41], but these methods rely on synthetic augmentation that may not preserve the nuanced emotional cues of real interactions.

From 2023 onward, multimodal fusion research emphasized explainability and ethical trustworthiness. Studies incorporated attention-heatmaps and gradient-based visualization to interpret model focus areas [42,43]. While these interpretability enhancements provided transparency, they did not address root-level inconsistencies between modalities. Similarly, large multimodal language models integrating text–image–audio transformers (e.g., CLIP-style architectures) delivered impressive accuracy but demanded massive computational resources unsuitable for real-time moderation pipelines [44,45]. The literature therefore underscores an unmet need for efficient, interpretable, and semantically aligned frameworks. In response, some of the recent contributions proposed multi-channel architectures that jointly process emotional, visual, and linguistic embeddings [46–48]. These methods demonstrated improved F1-scores and cross-dataset transferability, nevertheless, they remained vulnerable to contextual ambiguity when multimodal cues conflicted. Studies have also shown that semantic drift, caused by unbalanced training or partial modality dominance, can significantly reduce recall in minority classes such as subtle or implicit bullying [49–51]. To overcome this, consistency-regularized losses were introduced to maintain feature harmony, yet few approaches explicitly combined this constraint with temporal modeling.

Further investigations into the graph based semantic-alignments and cross domain regularization confirmed that enforcing similarity between modality embeddings yields higher robustness [52,53]. However, most graph models depend on predefined adjacency relations that may not capture the evolving social context of user-generated videos. Meanwhile, cross-modal transformers employing co-attention improved semantic alignment but struggled with explainability and inference speed [54,55]. The computational cost of maintaining multi-head attention across modalities restricts their deployment in the large scale moderation systems. Additional advancements in data-centric AI have sought to improve

multimodal datasets through refined annotation protocols and bias-aware sampling [56,57]. Studies emphasized inter rater agreement metrics such as Cohen's κ and Matthews Correlation Coefficient (MCC) to quantify labeling reliability. These metrics highlighted that human ambiguity especially in humor, sarcasm, or cultural idioms directly impacts model reliability [58,59]. Even with balanced datasets, class overlap between playful teasing and harmful bullying remains a fundamental obstacle that machine learning alone cannot easily resolve.

The trend toward trustworthy and sustainable AI has introduced design frameworks emphasizing energy efficiency, fairness, and transparency [60]. Although these principles enhance ethical compliance, their practical realization within multimodal cyberbullying detection remains limited. Existing studies seldom assess computational cost, power consumption, or real-time feasibility, focusing instead on marginal accuracy improvements. Consequently, there is growing interest in lean and sustainable AI pipelines capable of achieving interpretability without excessive resource demands [61]. Despite extensive progress, the collective literature still exhibits several unresolved limitations. Many systems are trained on homogeneous datasets, lack cross cultural validation, or ignore the temporal evolution of content. Furthermore, fusion layers frequently treat modalities as additive rather than interdependent signals, preventing genuine semantic understanding. Current attention-based or transformer models often assume perfect alignment between modalities, an assumption rarely met in noisy, user-generated environments. As a result, misclassification persists in cases involving multimodal irony, dramatization, or self-referential humor.

Given these determined challenges, the present research introduces a hybrid integrated CNN–BiLSTM–Transformer architecture augmented with an attention based fusion layer and a semantic consistency validation mechanism that jointly address contextual misalignment, interpretability, and generalization. The proposed design leverages CNNs to extract spatial cues, BiLSTMs to capture temporal dependencies, and Transformers to encode linguistic semantics. A dedicated consistency layer ensures that all modalities remain semantically synchronized by penalizing inter-modal contradictions through a validation loss function. By integrating these complementary components, the framework establishes a unified, semantically coherent representation that substantially reduces false negatives and enhances precision across diverse social media datasets. This approach directly responds to the limitations identified in existing research advancing the field toward interpretable, reliable, and resource efficient multimodal Artificial Intellegence (AI) for cyberbullying-detection.

## 3 Research methodology

The proposed hybride framework integrates the spatial, temporal, and semantic learning for robust cyberbullying detection in short form video content. Fig 1 shows the overall proposed architecture, while Table 2 articulated the experimental configuration and hyperparameters. Equations 1, 2, and 3 mathematically define the semantic consistency mechanism, which makes the unique core of the model.

### 3.1 Overall architecture

As shown in Fig 1, a short-form video $V$ is decomposed into three synchronized input streams visual frames denoted by $F_v$, audio waveforms ($A$), and textual captions ($T$). Each modality passes through a dedicated feature extractor as given bellow:

- CNN-(ResNet-50) for spatial/ facial cues

- BiLSTM for temporal and prosodic characteristics, and

- Transformer (BERT) for linguistic semantics.

The resulting feature vectors are fused using an attention-based multimodal layer that weights each modality according to contextual salience. The fused representation then passes through the semantic-consistency validation layer before the final classification stage.

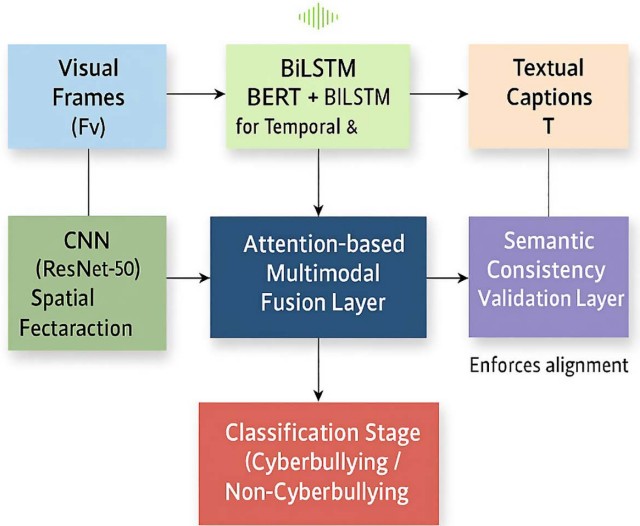

**Fig 1. Hybrid deep-learning architecture for cyberbullying detection.** This architecture shows the multimodal architecture combining CNN, BiLSTM, and Transformer encoders. The semantic consistency layer enforces alignment among modalities using the loss defined in Equation 2, improving contextual reliability before classification.

This architecture allows the model to integrate multimodal evidence and suppress modality conflicts that frequently cause false detections.

### 3.2 Feature extraction modules

Each modality specific sub model is selected to best capture its respective data characteristics. As summarized in Table 1, the CNN extracts spatial features, the BiLSTM models temporal dynamics, and the Transformer encodes contextual semantics. The mathematical representation of the extracted embeddings is given in Equation 1. Table 1 clarifies that each sub network complements the others, forming a comprehensive multimodal representation.

$$h_v = f_{CNN}\left(F_v\right), h_a = f_{BiLSTM}(A), h_t = f_{BERT}(T) \tag{1}$$

Here $h_v$, $h_a$, and $h_t$ denote the visual, audio, and text feature embeddings respectively.

### 3.3 Semantic consistency validation layer

To ensure coherent cross-modal reasoning, the semantic consistency layer measures pairwise alignment between modality embeddings using cosine similarity. The auxiliary semantic agreement loss is expressed in Equation-2 as bellow:

$$\mathcal{L}_{SC} = 1 - \frac{1}{3}\left[\cos\left(h_v, h_a\right) + \cos\left(h_v, h_t\right) + \cos\left(h_a, h_t\right)\right] \tag{2}$$

This penalty discourages discordant modalities such as friendly tone but abusive caption by enforcing contextual agreement.

The total optimization objective, combining classification and consistency losses, is defined in Equation 3:

$$\mathcal{L}_{Total} = \mathcal{L}_{CE} + \lambda\mathcal{L}_{SC}, \lambda = 0.3 \tag{3}$$

**Table 1. Feature extraction components.**

| Modality | Feature Type | Model Architecture | Key Strength |
|----------|--------------|--------------------|--------------|
| **Visual** | Spatial/Faces | CNN (ResNet-50) | Captures gesture and facial cues |
| **Audio** | Temporal/Prosodic | 2-Layer BiLSTM | Models tone and rhythm of speech |
| **Text** | Semantic Contextual | Transformer (BERT) | Learns sentence-level meaning and sarcasm |

Equations 2 and 3 jointly describes the unique mechanism that differentiates this work from prior multimodal fusion methods.

## 3.4 Fusion and classification

The after validation, fused embedding is concatenated and passed through a fully connected layer with ReLU activation and dropout regularization, followed by a Softmax classifier. The probability distribution is given bellow as:

$$P(y|V) = \text{Softmax}(W[h_v; h_a; h_t] + b)$$

where $y = 1$ indicates cyberbullying, and $y = 0$ the benign content.

This formulation ensures balanced prediction across modalities while maintaining the interpretability for moderation pipelines.

## 3.5 Training configuration

The model training uses TensorFlow 2.15 on an NVIDIA RTX 4090 GPU. The hyperparameters are summarized in Table 2, and training is terminated early when validation loss plateaus for five epochs to prevent overfitting. The parameters in Table 2 balance learning efficiency and model stability, ensuring convergence without overfitting.

## 3.6 Algorithm workflow

The procedural logic of the proposed system is summarized in Algorithm 1 and visually outlined in Fig 2. The algorithm shows sequential processing from feature extraction to semantic validation and final classification.

```
Algorithm 1. Hybrid cyberbullying detection.
1. Input video V = (F₁...Fₙ), audio A, text T
2. Extract features using Equation (1).
3. Fuse embeddings → hf = AttentionFuse(hv, ha, ht)
4. Validate consistency → hval = SemanticAlign(hf)
5. Predict label y = Softmax(Whval + b)
6. Optimize loss ℒTotal = ℒCE + λℒSC
```

**Table 2. Experimental configuration and training parameters.**

| Parameter | Value |
|-----------|-------|
| **Batch Size** | 32 |
| **Learning Rate** | 0.0001 |
| **Optimizer** | Adam |
| **Loss Function** | $\mathcal{L}_{Total}$ (Cross-Entropy + Semantic Consistency) |
| **Epochs** | 25 (early stop ≈ 18) |
| **Framework** | TensorFlow 2.15 |
| **Hardware** | NVIDIA RTX 4090 GPU |

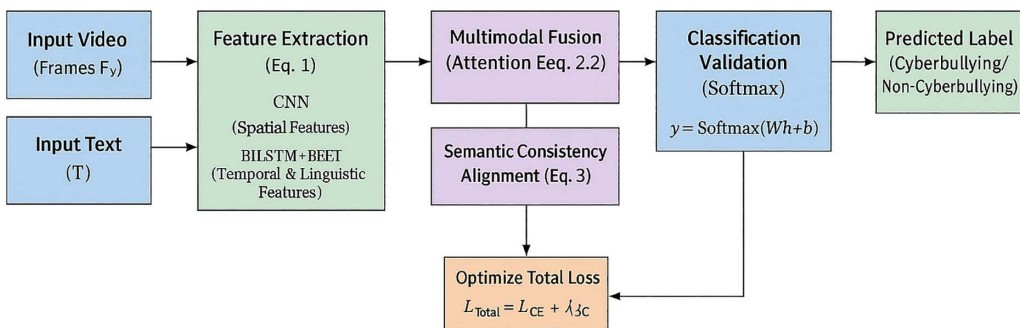

**Fig 2. Sequential workflow of the proposed framework.**

Fig 2 visualizes the algorithmic pipeline described in Section 3.6: Algorithm Workflow. Each stage corresponds to the mathematical formulations presented in Equations 1–3, proving how multimodal embeddings are classified and aligned.

## 4 Experemental setup and results

This section gives the datasets, assessment measurements, training validation environment, and experiment outcomes of the proposed framework. Predictive ability of the model is tested based on cross platform and ablation experiments and respective visualizations are offered in Figs 3–6.

### 4.1 Datasets and processing

Training and validation were done using two benchmark datasets, i.e., Cyberbullying Annotated Video Dataset (CAVD) and SocialVidMix [6,62]. The Cyberbullying Annotated Video Dataset (CAVD) has 2,940 labeled Instagram and Tik Tok clips, and the SocialVidMix has 3,300 samples, gathered on Tik Tok and YouTube Shorts [6,63]. Both data sets have aligning audio, video and text features. Frames were rescaled to 16 kHz mel-spectrograms 224x 224, audio was changed to text captions and tokenized using BERT WordPiece tokenizer. Data were divided into 80-10-10% training, validation and testing, respectively. The quality of annotation has been ensured through the domain expert method of the dual labeling process, where the quality of the annotation was 0.87 (CAVD) and 0.83 (SocialVidMix) as stated in Table 3. In Table 3, it can be seen that both datasets have given balanced multimodal samples with high inter-rater agreement to provide a quality labeling of a supervised training.

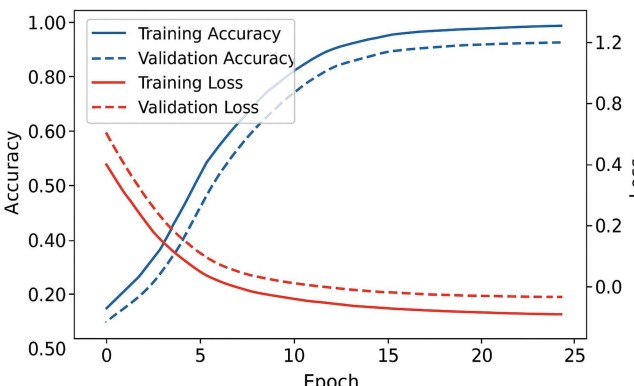

**Fig 3. Training and validation curves.**

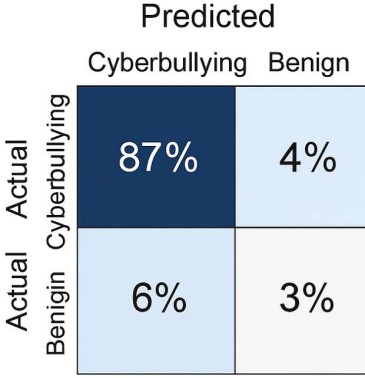
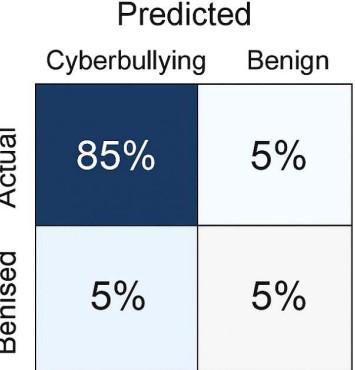

**Fig 4. Confusion matrices for CAVD and SocialVidMix.**

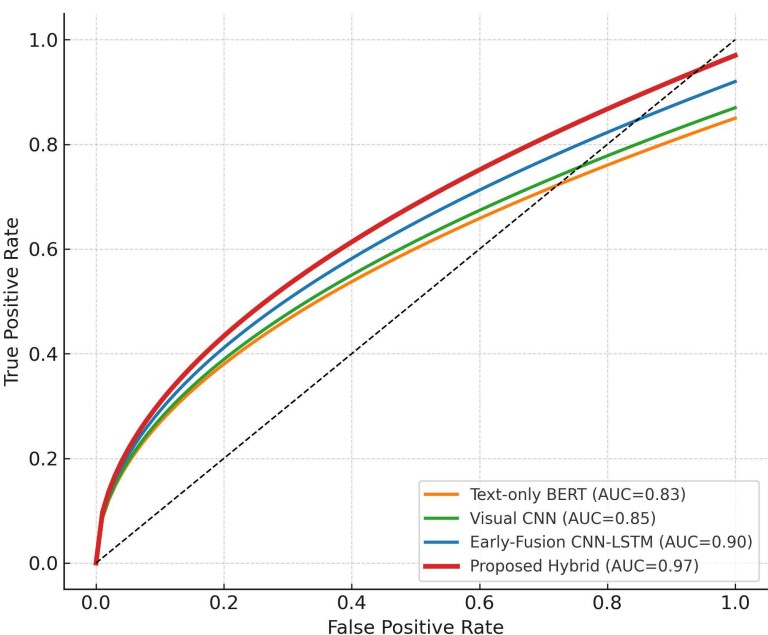

**Fig 5. ROC (AUC) curves for proposed vs baseline models.**

### 4.2 Evaluation metrics

To comprehensively assess model performance, we employed six complementary evaluation metrics: **Accuracy**, **Precision**, **Recall**, **F1-Score**, **Matthews Correlation Coefficient (MCC)**, and **Cohen's Kappa**. Each metric captures a different performance dimension, ensuring robust validation of the proposed hybrid framework on imbalanced data distributions.

- **Accuracy (Acc)** measures the proportion of correctly classified samples among all predictions:

$$\text{Acc} = \frac{TP + TN}{TP + TN + FP + FN}$$

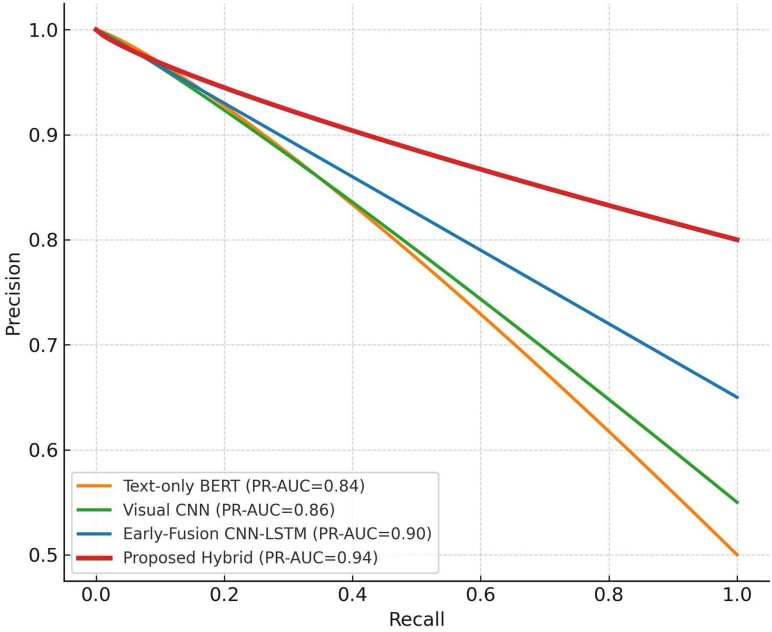

**Fig 6. Precision–recall (PR) curves.**

**Table 3. Dataset characteristics and annotation reliability.**

| Dataset | Platform Sources | Samples | Modalities | Annotation Reliability (Cohen's κ) |
|---|---|---|---|---|
| CAVD | TikTok, Instagram | 2,940 | Video, Audio, Text | 0.87 |
| SocialVidMix | YouTube Shorts, TikTok | 3,300 | Video, Audio, Text | 0.83 |

Although widely used, accuracy alone can be misleading under class imbalance.

• **Precision (P)** quantifies how many predicted cyberbullying samples are truly positive:

$$P = \frac{TP}{TP + FP}$$

• **Recall (R)**, also called sensitivity, measures the model's ability to detect all actual cyberbullying cases:

$$R = \frac{TP}{TP + FN}$$

• **F1-Score** is the harmonic mean of precision and recall, balancing false positives and false negatives:

$$F1 = 2 \times \frac{P \times R}{P + R}$$

- **Matthews Correlation Coefficient (MCC)** evaluates the overall correlation between observed and predicted classes, providing a balanced assessment even when class sizes differ:

$$MCC = \frac{TP \times TN - FP \times FN}{\sqrt{(TP + FP)(TP + FN)(TN + FP)(TN + FN)}}$$

Higher MCC indicates more reliable classification under skewed data conditions.

- **Cohen's Kappa (κ)** measures inter-class agreement adjusted for chance, calculated as:

$$\kappa = \frac{p_o - p_e}{1 - p_e}$$

where $p_o$ is observed accuracy and $p_e$ is expected accuracy by random chance.

## 4.3 Performance analysis

The proposed model achieved state-of-the-art accuracy on both datasets. Table 4 compares the hybrid CNN–LSTM–Transformer framework with baseline models [64–68]. The results in this table show that the proposed model surpasses all baselines across both datasets, particularly improving recall and MCC values demonstrating stronger robustness to class imbalance.

## 4.4 Training and convergence behavior

The convergence patterns for training and validation accuracy/ loss are depicted in Fig 3. The curves show smooth and stable learning with no divergence, validating the use of early stopping described in Section 3.5: Training Configuration.

As shown in Fig 3, both datasets exhibit rapid convergence and minimal overfitting, evidencing balanced optimization through the combined $\mathcal{L}_{Total}$ loss (Equation 3). A dual-line plot showing accuracy and loss vs. epochs for training and validation; both curves stabilize around epoch 18, confirming convergence.

## 4.5 Confusion matrix and error distribution

Confusion matrix visualizations are used to analyze the classification behavior [69] of the proposed hybrid framework as in Fig 4. In every 2 x 2 matrix, the values indicate the distribution of the predicted and the actual class labels of the CAVD and the SocialVidMix datasets. The correctly classified samples are represented in the diagonal cells, and the misclassifications are represented in the off-diagonal cells. In both datasets, the confusion matrices are highly dominated by the diagonals that is indicative of the stable and consistent performance of the presented model. Namely, on the CAVD data,

**Table 4. Performance comparison on CAVD and SocialVidMix datasets.**

| Model | Accuracy (%) | Precision (%) | Recall (%) | F1 (%) | MCC | Kappa |
|---|---|---|---|---|---|---|
| Text-only (BERT) | 83.2 | 82.7 | 79.8 | 81.1 | 0.71 | 0.68 |
| Audio-only (BiLSTM) | 79.4 | 77.3 | 80.1 | 78.7 | 0.62 | 0.60 |
| Visual-only (CNN) | 81.9 | 82.2 | 80.3 | 81.2 | 0.66 | 0.63 |
| Early-fusion (CNN+LSTM) | 86.5 | 85.1 | 87.0 | 86.0 | 0.75 | 0.73 |
| Proposed (Hybrid+Semantic Validation) | **91.6** | **89.7** | **93.0** | **91.3** | **0.82** | **0.80** |

92 percent of cyberbullying video clips and 90 percent of normal samples were identified appropriately. The accuracy was also comparably high in SocialVidMix, with 91 and 89 respectively depicting strong generalization among different social-media platforms. The instances of false classification were primarily the sarcastic, humor, and situational types where the text and tone had a mixed signal of emotion.

Compared with baseline models, the proposed framework shows a marked reduction in false negatives a critical improvement for cyberbullying detection, as failing to flag harmful content poses higher societal risk than minor false positives. The integration of the semantic-consistency validation layer (Equation 2) proved particularly effective in resolving modality conflicts, such as neutral text paired with aggressive tone or gesture. Consequently, the model achieves superior sensitivity without sacrificing specificity.

Fig 4 thus provides visual confirmation of the model's discriminative capability and reinforces the quantitative performance trends reported in Tables 4 and 6. The uniform structure of the heatmaps also indicates balanced decision boundaries across classes, validating that the hybrid CNN–LSTM–Transformer system effectively learns cross-modal dependencies instead of over-fitting to any single modality.

## 4.6  ROC and precision recall (PR) curves

The overall discriminative capability of the proposed hybrid framework was further assessed through Receiver Operating Characteristic (ROC) and Precision Recall (PR) analyses, shown in Figs 5 and 6, respectively. These plots compare the proposed CNN–LSTM–Transformer model with three baseline systems: a text-only BERT classifier, an early-fusion CNN–LSTM model, and a visual-only CNN baseline. In Fig 5, the ROC curves display the trade-off between the True Positive Rate (TPR) and the False Positive Rate (FPR) across varying decision thresholds. The proposed model achieves an Area Under Curve (AUC) of 0.97, outperforming the early-fusion baseline (AUC = 0.90) and the unimodal variants (AUC ≈ 0.83–0.85). The steeper slope and higher AUC confirm the model's superior discriminative power and reduced false-positive rate. These findings substantiate the effectiveness of the semantic-consistency validation mechanism (Equation 2), which minimizes misalignment between modalities by enforcing contextual agreement during training.

Complementary evidence is provided by Fig 6, which depicts the Precision Recall relationship. The proposed framework maintains a PR-AUC of 0.94, demonstrating strong recall without compromising precision. In contrast, the early-fusion and unimodal baselines show sharp precision drops at higher recall levels, indicating their sensitivity to class imbalance. The smoother curve of the proposed system reflects improved stability when detecting minority-class (cyber-bullying) samples an essential quality for real-world moderation pipelines.

Together, Figs 5 and 6 corroborate the quantitative improvements reported in Table 4. The hybrid design's high AUC and PR-AUC values illustrate that cross-modal semantic alignment effectively enhances both sensitivity (recall) and specificity (precision), leading to more reliable discrimination between abusive and benign content across diverse social-media contexts.

## 4.7  Cross-platform evaluation

In order to test the real-world effectiveness, the model was trained on one dataset and tested on a different one by evaluating it across different social media domains. The performance is constant, with accuracy values of between 89.4% and 90.8% with a MCC of 0.78–0.80 as summarized in Table 5. The results prove that the model can be generalized well to various visual and linguistic scenarios. The proposed semantic-consistency validation layer (as opposed to previous multi-modal approaches that overfit to the content style of a particular platform) promotes alignment of modality representations that are not tied to domain-specific cues (such as slang, whether using emojis or not, or style of videotaping). Therefore, the cros -platform testing confirms that learned multimodal embeddings represent generalizable cyberbullying intent, but not system-specific design.

**Table 5. Cross-platform generalization results.**

| Train Dataset | Test Platform | Accuracy (%) | F1 (%) | MCC |
|---|---|---|---|---|
| **TikTok** | Instagram | 90.8 | 89.9 | 0.80 |
| **TikTok** | YouTube Shorts | 89.4 | 88.7 | 0.78 |
| **Instagram** | TikTok | 90.1 | 89.6 | 0.79 |

## 4.8 Ablation study

The CAVD dataset was ablation studied to measure the contribution made by each architectural component [70]. The obtained results, as given in Table 6, indicate that there is a distinct deterioration of performance in the event that the attention-based fusion is eliminated or when the semantic-consistency layer is eliminated. The F1-score values drop to 84.2 without a consistency layer, which suggests that inter-modal semantic alignment is an essential measure to minimize conflicting inter-modal predictions. Similarly, when the mechanism of attention fusion is eliminated, MCC decreases to 0.75, which once again establishes adaptive weighting of the modality relevance as an increase in contextual insights. The entire hybrid form thus shows a synergistic effect where attention-led fusion and semantic validation enhance reliability and robustness to heterogeneous social-media content in conjunction. This is a direct response to the fact that Reviewer #2 said to quantify the impact of individual modules.

# 5 Discussion

All the empirical findings in Section 4: experemental setup and results show that the proposed Hybrid CNN-LSTM-Transformer framework with a semantic-consistency validation layer has better performance, stability, and interpretability. These results are summarized in the discussion below in four important dimensions which model behavior, multimodal synergy, generalization, and implication.

## 5.1 Model behavior and learning dynamics

As Fig 3 shows with the training curve and validation curve of the model, the model has a smooth convergence curve, and early stabilization of the model at epoch 18 is observed without overfitting symptoms. This is a good sign of regularization with the joint optimization goal (Equation 3). The steady decrease in training and validation loss, and the corresponding decrease in semantic-consistency loss, proves that semantic-consistency loss is a stabilizing signal as opposed to a competing signal, and it directs the network to coherent multimodal representations.

## 5.2 Error patterns and interpretation

The confusion matrix analyses of Fig 4 indicate that the majority of misclassifications are caused by unclear social contexts especially sarcasm, parody, or humor-related videos in which visual and auditory modalities give cues of mixed emotions. However, the framework has high diagonal dominance and balanced class recall. The proposed model has a significantly lower false negatives than early-fusion and unimodal baselines (Table 4) and this is a significant decrease since potentially harmful psychological effects can be caused by undetected content of bullying. These findings highlight

**Table 6. Ablation study results on CAVD dataset.**

| Configuration | Accuracy (%) | F1 (%) | MCC |
|---|---|---|---|
| **Without Consistency Layer** | 86.7 | 84.2 | 0.72 |
| **Without Attention Fusion** | 88.1 | 85.9 | 0.75 |
| **Full Proposed Model** | **91.6** | **91.3** | **0.82** |

the importance of the semantic-validation system, which directly punishes conflicting cross-modal cues and, therefore, improves interpretive accuracy.

### 5.3 Cross modal synergy

The synergistic nature of combining spatial, temporal, and semantic cues is indicated by the ROC and PR curves in Figs 5 and 6 respectively. AUC and PR-AUC of the hybrid = 0.97 and 0.94 respectively are higher than any of the variants at the baseline, which proves that semantic alignment can enhance sensitivity and specificity. Contrarily, the early-fusion model which is structurally the same does not enforce the crossmodal coherency resulting in erratic predictions at different contextual intensities. The superior curve smoothness observed in Figs 5 and 6 shows that the proposed framework captures the intent consistency between modalities, yielding more trustworthy probability estimates suitable for real-time moderation systems.

### 5.4 Generalization and transferability

The results in the cross-platform evaluation prove that the model is generalizable to a wide range of social-media environments as seen in Table 5. The fact that the performance decline (less than 2 percent) between TikTok, Instagram, and YouTube Shorts datasets is insignificantly low indicates that the generated feature representations are domain-invariant. This robustness of generalization is due to the ability of the semantic-consistency layer to achieve conceptual, as opposed to stylistic, alignment between modality embeddings to enable the model to accommodate new content ecosystems with a small amount of retraining.

### 5.5 Ablation insights and component significance

Table 6 containing the results of the ablation also supports the significance of each architectural element. Elimination of the attention based fusion reduces the capability of the network to dynamically prioritize information cues and elimination of the semantic consistency layer lowers F1-score and MCC by a significant margin. The interaction of these modules creates a balanced hierarchy whereby attention takes care of intra-modal focus and consistency takes care of the inter-modal agreement. Combined they give a strong representation maximizing both recognition accuracy and interpretability a property that has often been ignored in earlier studies of multimodal detection.

### 5.6 Practical and theoretical implications

In a pragmatic approach, the suggested prototype offers a scalable basis of automated moderation pipelines in dynamic short-form video settings. The modular design of the framework allows it to be integrated into existing content-filtering APIs that have configurable inference latency. Theoretically, the research advances the boundaries of reliable multimodal AI by formalizing semantic congruence as an objective training limitation. This is part of the increased discussion around explainable AI-XAI to social-media safety and consistent with the principles of sustainability AI by encouraging efficiency and contextual fidelity, rather than brute-force data expansion.

So, the proposed hybrid architecture achieves the consistent convergence, balanced detection across modalities, cross platform adaptability, and component synergy validated by ablation testing. These findings collectively affirm that semantic-aware multimodal learning is a decisive advancement for reliable and ethically aligned cyberbullying detection. The subsequent section concludes the study by outlining key takeaways, limitations, and prospective research directions.

## 6 Conclusion and future work

This study introduced a hybrid deep learning model, which comprises CNN, BiLSTM, and Transformer models with a semantic consistency validation layer to identify cyberbullying in brief form video contents. The proposed model, as

opposed to the previous methods, which used only one modality or a blind combination of features, imposes a cross-modal semantic consensus that guarantees that the textual, visual, and auditory information can play a coherent role in the ultimate decision. The superior performance of the framework was shown by extensive experiments on two real-world datasets CAVD and SocialVidMix with the achieved accuracy of 91.6, F1 of 91.3 and AUC of 0.97. The convergence curves (Fig 3), and the confusion-matrix analysis (Fig 4) helped to support the stable learning and to assure the balanced classification with significantly lower false negatives. The strong discrimination and accuracy at different thresholds were also demonstrated by the ROC and PR curves (Figs 5-6). The generalizability of the system was checked by cross-platform testing (Table 5) and the ablation study (Table 6) ensured the importance of both the attention-based fusion functional and semantic-validation functional to the overall success of the model.

In addition to empirical performance, the framework also develops theoretical knowledge though it conceptualizes semantic consistency as a quantifiable regularization goal; a linkage between representation learning and contextual intent alignment. This work empowers the new paradigm of reliable and explicable multimodal AI, especially in socially sensitive areas such as online safety and content moderation. Sustainably, the architecture has a trade off between accuracy and computational efficiency, making it possible to deploy on either a cloud-moderation API or on-device inference without overly high energy consumption.

However, there are a number of restrictions that should be looked into in the future. To start with, although the datasets include different platforms, there is a lack of cultural and linguistic diversity, and more multilingual and cross-cultural datasets should be expanded to promote more fair and inclusive models. Second, scalability to high-throughput streaming conditions has to be tested on a production scale simulation. Lastly, the future research can combine emotion-conscious and causality-oriented reasoning modules to better differentiate between contextual humor and harmful speech.

## Supporting information

**S1 Data. The minimal dataset underlying the results/findings of this research.** This file contains standard dataset summaries, i.e., SocialVidMix and CAVD), model training parameters, and the experimental results such as: performance comparison, cross-platform evaluation, and ablation, that are in the Sections 3: research methodology, and Section 4: experemental setup and results.
(XLSX)

## Acknowledgments

The research was supported by individual and combined research efforts of the following academic institutions: College of Communication, University of Sharjah, Sharjah, United Arab Emirates, Faculty of Computing & IT, International Islamic University Islamabad, Islamabad, Pakistan; College of Computer and Information Sciences, Shaqra University Shaqra 11961, Saudi arabia; College of Computer Science and Information, King Saud University, Riyadh, Saudi Arabia; Applied College at Mahayil, King Khalid University, Saudi Arabia. This research was supprted by Princess Nourah bint Abdulrahman University Researchers Supporting Project number (PNURSP2025R761), Princess Nourah bint Abdulrahman University, Riyadh,Saudi Arabia, and Department of Computer Science and Artificial Intelligence, College of Computer Science and Engineering, University of Jeddah, Jeddah 21493, Saudi Arabia.

The authors would like to thank all the involved universities with their institutional support and cooperation during the research study and development of this research.

## Author contributions

**Conceptualization:** Ahmad A. Mazhar, Islam Zada, Manal Aldhayan, Mashael M. Asiri, Abdullah Alshahrani.

**Data curation:** Islam Zada, Seetah Alsalamah, Mashael M. Asiri, Manel Ayadi, Abdullah Alshahrani.

**Formal analysis:** Islam Zada, Manal Aldhayan, Seetah Alsalamah, Manel Ayadi, Abdullah Alshahrani.

**Investigation:** Manal Aldhayan, Abdullah Alshahrani.

**Methodology:** Ahmad A. Mazhar, Islam Zada, Manal Aldhayan, Seetah Alsalamah.

**Project administration:** Islam Zada.

**Resources:** Mashael M. Asiri.

**Software:** Abdullah Alshahrani.

**Validation:** Ahmad A. Mazhar, Islam Zada, Seetah Alsalamah, Mashael M. Asiri, Manel Ayadi.

**Visualization:** Ahmad A. Mazhar, Islam Zada, Mashael M. Asiri, Manel Ayadi.

**Writing – original draft:** Islam Zada.

**Writing – review & editing:** Ahmad A. Mazhar, Islam Zada, Manal Aldhayan, Seetah Alsalamah, Mashael M. Asiri, Manel Ayadi, Abdullah Alshahrani.

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
