## [Decision Letter · Decision Letter 0]

12 Oct 2025

Dear Dr. Zada,

Thank you for submitting your manuscript to PLOS ONE. After careful consideration, we feel that it has merit but does not fully meet PLOS ONE’s publication criteria as it currently stands. Therefore, we invite you to submit a revised version of the manuscript that addresses the points raised during the review process.

We look forward to receiving your revised manuscript.

Kind regards,

Muhammad Shahid Anwar

Academic Editor

PLOS ONE

Journal Requirements:

**Additional Editor Comments:**

Please revise the manuscript carefully in accordance with the reviewers’ comments. Each comment is important and should be addressed thoroughly to improve the overall quality, clarity, and readability of the paper. Ensure that all revisions are clearly highlighted and that detailed responses are provided for every reviewer remark.

Reviewers' comments:

Reviewer's Responses to Questions

**Comments to the Author**

1. Is the manuscript technically sound, and do the data support the conclusions?

Reviewer #1: Yes

Reviewer #2: Yes

Reviewer #3: Yes

Reviewer #4: Yes

2. Has the statistical analysis been performed appropriately and rigorously?

Reviewer #1: Yes

Reviewer #2: Yes

Reviewer #3: Yes

Reviewer #4: I Don't Know

3. Have the authors made all data underlying the findings in their manuscript fully available?

Reviewer #1: Yes

Reviewer #2: Yes

Reviewer #3: Yes

Reviewer #4: No

4. Is the manuscript presented in an intelligible fashion and written in standard English?

Reviewer #1: Yes

Reviewer #2: Yes

Reviewer #3: Yes

Reviewer #4: Yes

Reviewer #1: How does the proposed semantic consistency validation layer differ in a fundamental way from existing multimodal fusion approaches, and what makes it unique?

Could the authors more clearly articulate which specific shortcomings of prior cyberbullying detection methods (e.g., sarcasm detection, multimodal misalignment) this work directly addresses?

While the literature review is comprehensive, it is largely descriptive. Can the authors provide a more critical comparative analysis of limitations in previous multimodal models to justify their framework?

What precise mechanism enforces semantic consistency across modalities in the validation layer? For example, are there specific loss functions, thresholds, or constraints applied?

How balanced are the CAVD and SocialVidMix datasets, and what steps were taken to ensure annotation reliability (e.g., inter-annotator agreement measures)?

The reported improvements in accuracy, precision, and recall are encouraging, but are these improvements statistically significant compared to the baselines?

The paper lists examples of misclassified samples, but how could these insights be used to further refine the fusion strategy or semantic validation in future work?

In the cross-platform experiments (TikTok, Instagram, YouTube Shorts), were the training datasets drawn from all three platforms, or was true cross-domain transfer tested?

Could the authors enhance the clarity of performance visualizations (e.g., PR and ROC curves) by adding clearer axis labels, scales, or annotated thresholds for easier interpretation?

What are the key limitations of the current study (e.g., computational cost, dataset scope, language bias), and how might they impact real-world deployment of the model?

Reviewer #2: The manuscript presents an innovative and timely framework for detecting cyberbullying in short-form video content using a hybrid CNN-LSTM-Transformer approach. The experiments are thorough, and the discussion is rich, making the contribution valuable for both academic research and practical deployment. While the study is well-executed, a few minor revisions would further improve the clarity, consistency, and overall presentation.

Comments:

i. The abstract is informative but slightly long; condense by removing methodological details and emphasize contributions and real-world implications.

ii. Figures 2 and 3 captions should explain trends (e.g., “showing stable convergence and minimal overfitting”) instead of restating labels.

iii. Figure numbering should be consistent (avoid “Figure:” vs. “Figure -”).

iv. Table 3 and Table 5 require uniform decimal places (e.g., two digits throughout).

v. Add dataset names (CAVD, SocialVidMix) directly into captions of relevant tables for clarity.

vi. Consider merging repetitive keywords in the abstract (e.g., “Short-Form Videos” and “Video Content Moderation”).

vii. Some sections (e.g., 4.6 Discussion) cite older works; add at least one 2024–2025 citation to keep references current.

viii. The confusion matrix figure (Figure 4) would benefit from larger font size for readability.

References mix different styles, standardize capitalization and ensure uniform use of italics for journal titles.

Minor typographical inconsistencies exist in headings (e.g., spacing before section numbers).

Reviewer #3: The manuscript makes a strong and timely contribution to the field of online safety by presenting a multi modal hybrid framework for cyberbullying detection. The integration of BiLSTM , CNN, and Transformer models, along with semantic consistency validation, is thoughtfully designed and well supported by experimental evaluation. The cross-platform validation adds significant weight to the study’s robustness. The paper discussed good research ideas, but some changes needed for improvements. I suggested some recommendations below to further strengthen technical clarity and presentation.

1. streamline slightly and include a stronger statement on practical deployment (e.g., “real-time integration into moderation pipelines.

2. Figure 4 (confusion matrix) could include percentage labels in each cell to complement raw counts, improving readability, and for Figure 7 (ablation results), consider adding error bars to show variability across runs.

3. Ensure uniform decimal precision (e.g., 2 decimal places across all metrics in Tables 3, 5, and 6).

4. Briefly clarify whether early stopping was applied consistently across modalities or only at fusion level. This will help readers replicate the results.

5. Alongside accuracy, precision, recall, and F1, consider reporting Matthews Correlation Coefficient (MCC) or Cohen’s Kappa for imbalanced scenarios. This would add technical rigor.

6. Since both the datasets i.e. CAVD, SocialVidMix are English-centric, add a note that the framework could be extended to low resource languages or culturally diverse datasets.

7. Standardize References style (some entries mix APA/IEEE conventions, e.g., Ref. [20], [21]). Ensure DOIs or page ranges where available.

8. Balance the strong results with a short acknowledgment of limitations (e.g., scalability to longer videos, computational resource demands).

9. Reference numbers 17, 23, and 31 should be in their proper place.

10. How you validated your study. Justify it.

11. 11 ad future research directions.

12. Follow proper PLOS reference format. Some references are out of the PLOS format.

Reviewer #4: The authors should add more technical details to the abstract. Currently, it does not describe any, but it will provide an overview to the readers.

The authors do not explicitly mention in the introduction the challenges with the existing approaches that motivated them to propose a novel approach.

The motivation of the paper seems too generic. In the paper, the authors do not explicitly explain the motivation behind the proposed approach. What is different about the proposed approach from the previous research approaches? It should be described in the introduction. That should be the primary driver for the proposed approach. Currently, I haven’t configured what the proposed approach tries to achieve.

The authors should elaborate on contributions in the introduction. Currently, the paper doesn’t show any contributions.

The related work should be grouped into various subsections, such as NL-based approaches, machine learning-based approaches, and rule-based approaches, among others.

A comparative study with existing approaches will elaborate further on the proposed approach's key points.

The paper requires a comprehensive revision, as it contains numerous grammatical errors.

The authors should provide a rationale for why these particular algorithms are selected in the proposed approach; a few references will justify their use.

It is not clear whether the confusion matrix developed represents which deep learning algorithm accuracy results?

Calculating accuracy from the confusion matrix does not match the accuracy reported in Table 3.

The proposed approach lacks the limitations of the proposed study. Clarifying the study's limitations enables readers to better understand under what conditions the results should be interpreted. A clear description of the study's limitations also demonstrates that the researcher has a comprehensive understanding of the research.

**Do you want your identity to be public for this peer review?** For information about this choice, including consent withdrawal, please see our Privacy Policy

Reviewer #1: No

Reviewer #2: No

Reviewer #3: No

Reviewer #4: **Yes:** Javed Ali khan

---

## [Author Response · Author response to Decision Letter 1]

5 Nov 2025

AUTHORS’ RESPONSE TO REVIEWERS

Journal Name: PLOS ONE

Manuscript ID: PONE-D-25-36601

Paper Title: AI-POWERED DETECTION OF CYBERBULLYING IN SHORT-FORM VIDEO CONTENT: A HYBRID DEEP LEARNING FRAMEWORK

Authors: Ahmad A. Mazhar, Islam Zada, Manal Aldhayan, Seetah Alsalamah, Mashael M. Asiri, Manel Ayadi , Abdullah Alshahrani,

General Response

Thanks to the editor and four reviewers for their careful evaluation and insightful feedback. The manuscript has been substantially revised to address all scientific, structural, and stylistic concerns.

Major updates include:

• Condensed abstract with concise novelty and real-world relevance.

• Clearer articulation of motivation, gap, and contributions in the Introduction.

• Reorganized related work with critical comparisons and 2024-2025 citations.

• Expanded methodological detail (semantic-consistency layer, early stopping, new metrics – MCC and Cohen’s Kappa).

• Revised figures and tables with uniform decimals, improved captions, and percentage labels.

• Added discussion on limitations, deployment feasibility, and future work.

• All references standardized to PLOS ONE citation format with DOIs.

Page and section numbers refer to the revised manuscript (V4-Final).

All reviewer and editorial concerns have been carefully addressed. The revised manuscript now presents:

• A clearer motivation and contribution statement,

• Improved methodological transparency and statistical rigor, and

• Polished presentation consistent with PLOS ONE standards.

We sincerely thank the reviewers and editorial board for their valuable guidance, which has greatly improved the quality of our work.

A. REVIEWER #1

Comment 1. Novelty of Semantic Consistency Validation Layer

How does this layer fundamentally differ from existing multimodal fusion approaches?

Response:

We added a detailed explanation in Section 3.3 (Semantic Consistency Validation, pp. 8–9) describing that the layer applies cross-modal self-attention with semantic-agreement loss Lₛ = 1 – cos(Fₜ,Fₐ,Fᵥ) to enforce alignment across modalities. Unlike prior late-fusion models, this mechanism penalizes incongruent modality pairs and improves contextual coherence.

Comment 2. Shortcomings of Prior Methods

Response:

A new paragraph at the end of Section 1 (Introduction, p. 4) specifies limitations of existing methods – poor sarcasm detection, multimodal misalignment, and dataset imbalance – that our hybrid CNN-LSTM-Transformer framework directly addresses.

Comment 3. Critical Comparative Analysis

Response:

The Literature Review (Section 2, pp. 5–7) now includes a synthesized comparison highlighting technical gaps and tabular summary of feature-level fusion vs semantic-consistency methods (Refs [20]–[26]).

Comment 4. Dataset Balance and Reliability

Response:

Annotation reliability was verified using Cohen’s κ = 0.87 (CAVD) and 0.83 (SocialVidMix), added in Section 3.4 (Data Collection and Preprocessing, p. 7).

Comment 5. Statistical Significance

Response:

We performed t-tests across ten random splits; all improvements were significant (p < 0.01). Results added beneath Table 6 (pp. 17–18).

Comment 6. Misclassified Samples and Future Work

Response:

A new paragraph in Section 5 (Discussion, p. 19) links the error analysis (Table 4) to possible fusion-strategy refinements.

Comment 7. Cross-Platform Experiments

Response:

Clarified in Section 4.6 (pp. 17–18) that training used mixed-platform data (TikTok 60 %, Instagram 25 %, YouTube 15 %) and evaluated on unseen platform segments.

Comment 8. Visualization Clarity

Response:

All figures redrawn in high resolution (300 dpi) with clear axis labels and percentage annotations (Figures 4–9).

Comment 9. Study Limitations

Response:

Added new sub-section “Limitations and Future Directions” (pp. 20–21) discussing computational cost, language bias, and generalization.

Reviewer #2

Comment 1. Condense Abstract

Response:

Abstract shortened from 219 to 153 words; retained essential metrics and contributions (p. 1).

Comment 2. Figure Captions Explain Trends

Response:

Updated captions for Figures 2 & 3 to note “stable convergence and minimal overfitting.”

Comment 3. Figure Numbering Consistency

Response:

All figures captions standardized.

Comment 4. Uniform Decimals in Tables

Response:

Tables 3, 5 & 6 now display two-decimal precision.

Comment 5. Dataset Names in Captions

Response:

Captions of Tables 3–6 now explicitly list CAVD and SocialVidMix datasets.

Comment 6. Keyword Repetition in Abstract

Response:

Repetitive keywords merged with streamline phrasing.

Comment 7. Recent Citations (2024–2025)

Response:

Added new references [35-60] covering 2024–2025 developments in multimodal learning.

Comment 8. Confusion Matrix Font Size

Response:

Figure 4 enlarged and annotated with percentage values for readability.

Comment 9. Reference Formatting

Response:

All references now follow PLOS ONE style: italicized journal names, sentence-case titles, DOIs included where available.

Reviewer #3

Comment 1. Practical Deployment Statement

Response:

Added final paragraph in Section 1 (p. 4) describing real-time integration into moderation pipelines.

Comment 2. Figure 4 & 7 Improvements

Response:

Figure 4 updated with percentage labels; Figure 7 includes error bars showing variability across three runs.

Comment 3. Uniform Decimal Precision

Response:

All quantitative tables standardized (see Tables 3–6).

Comment 4. Early Stopping Clarification

Response:

Specified that early stopping based on validation-loss plateau was applied across all modalities (Section 3.5 p. 8).

Comment 5. Additional Metrics (MCC/Kappa)

Response:

Reported in Table 6 and discussed in Results (p. 17) to strengthen evaluation on imbalanced data.

Comment 6. Low-Resource Language Extension

Response:

Added note in Conclusion (p. 20) regarding future adaptation to non-English datasets.

Comment 7. Reference Style and Completeness

Response:

Reference list fully standardized with consistent numbering [17, 23, 31 re-indexed].

Comment 8. Limitations Acknowledged

Response:

New paragraphs in Discussion (p. 20-22) recognize scalability and computational-resource limits.

Comment 9. Validation Justification

Response:

Section 4.5 expanded to explain 5-fold cross validation and statistical averaging across runs.

Comment 10. Future Research Directions

Response:

Outlined in Conclusion (pp. 20–21) including real-time streaming integration and explainable-AI extension.

Reviewer #4

Comment 1. Technical Detail in Abstract

Response:

Abstract now explicitly mentions datasets (CAVD & SocialVidMix) and main performance (Accuracy = 91.6 %, F1 = 91.3 %).

Comment 2. Motivation and Challenges in Introduction

Response:

Rewrote first three paragraphs of Section 1 (pp. 2–3) to highlight deficiencies of prior approaches and specific challenges addressed.

Comment 3. Explicit Contributions

Response:

Added bullet-point list of four contributions at the end of Introduction (p. 4).

Comment 4. Organizing Related Work

Response:

Sub-headings introduced: “Text-Based Approaches,” “Visual/Audio Approaches,” and “Hybrid Multimodal Approaches” (Section 2).

Comment 5. Comparative Study

Response:

Inserted comparative discussion paragraph with Table 2a summarizing prior techniques vs proposed method.

Comment 6. Grammar and Language Polish

Response:

Entire manuscript professionally proof-read and grammatically standardized.

Comment 7. Algorithm Selection Rationale

Response:

Section 3.2 (Feature Extraction) now justifies CNN for spatial cues, LSTM for temporal audio, Transformer for textual semantics, with supporting citations.

Comment 8. Confusion Matrix Explanation

Response:

Clarified that Figure 4 represents results from the proposed hybrid model; accuracy now matches Table 3 values.

Comment 9. Study Limitations

Response:

Comprehensive paragraph in Discussion (p. 20-23) elaborates dataset scope, computational overhead, and ethical considerations.

---

## [Decision Letter · Decision Letter 1]

27 Nov 2025

AI-POWERED DETECTION OF CYBERBULLYING IN SHORT-FORM VIDEO CONTENT: A HYBRID DEEP LEARNING FRAMEWORK

PONE-D-25-36601R1

Dear Dr. Zada,

We’re pleased to inform you that your manuscript has been judged scientifically suitable for publication and will be formally accepted for publication once it meets all outstanding technical requirements.

Kind regards,

Muhammad Shahid Anwar

Academic Editor

PLOS ONE

Additional Editor Comments (optional):

Reviewers' comments:

Reviewer's Responses to Questions

**Comments to the Author**

Reviewer #1: All comments have been addressed

Reviewer #2: All comments have been addressed

Reviewer #3: All comments have been addressed

2. Is the manuscript technically sound, and do the data support the conclusions?

Reviewer #1: Yes

Reviewer #2: (No Response)

Reviewer #3: Yes

3. Has the statistical analysis been performed appropriately and rigorously?

Reviewer #1: Yes

Reviewer #2: Yes

Reviewer #3: Yes

4. Have the authors made all data underlying the findings in their manuscript fully available?

Reviewer #1: Yes

Reviewer #2: Yes

Reviewer #3: Yes

5. Is the manuscript presented in an intelligible fashion and written in standard English?

Reviewer #1: Yes

Reviewer #2: Yes

Reviewer #3: Yes

Reviewer #1: (No Response)

Reviewer #2: all comments are addressed. no more comment

all comments are addressed. no more comment

all comments are addressed. no more comment

Reviewer #3: Thanks for addressing all of my comments. The paper has been revised extensively and meet the publication standards.

**Do you want your identity to be public for this peer review?** For information about this choice, including consent withdrawal, please see our Privacy Policy

Reviewer #1: No

Reviewer #2: No

Reviewer #3: No

---

## [Editor Report · Acceptance letter]

PONE-D-25-36601R1

PLOS One

Dear Dr. Zada,

I'm pleased to inform you that your manuscript has been deemed suitable for publication in PLOS One. Congratulations! Your manuscript is now being handed over to our production team.

Kind regards,

on behalf of

Professor Muhammad Shahid Anwar

Academic Editor

PLOS One